# CERTIFIED *PEFTSmoothing*: PARAMETER-EFFICIENT FINE-TUNING WITH RANDOMIZED SMOOTHING

## ABSTRACT

Randomized smoothing is the primary certified robustness method for accessing the robustness of deep learning models to adversarial perturbations in the $l_2$-norm, by taking a majority vote over the multiple predictions of a random Gaussian perturbed input of the base classifier. To fulfill the certified bound and empirical accuracy of randomized smoothing, the base model either needs to be retrained from scratch to learn Gaussian noise or adds an auxiliary denoiser to eliminate it. In this work, we propose *PEFTSmoothing*, which teach the base model to learn the Gaussian noise-augmented data with Parameter-Efficient Fine-Tuning (PEFT) methods in both white-box and black-box settings. This design is based on the intuition that large-scale models have the potential to learn diverse data patterns, including the noise data distributions. In addition, we explore the possibility of combining *PEFTSmoothing* with the fine-tuning for downstream task adaptation, which allows us to simultaneously obtain a robust version of the large vision model and its adaptation tailored to downstream datasets. Extensive results demonstrate the effectiveness and efficiency of *PEFTSmoothing*, which allow us to certify over 98% accuracy for ViT on CIFAR-10, 20% higher than SoTA denoised smoothing, and over 61% accuracy on ImageNet which is 30% higher than CNN-based denoiser and comparable to the Diffusion-based denoiser.

## 1 INTRODUCTION

Certified robustness is the primary method to evaluate the robustness of deep learning systems to adversarial perturbations within specific bound (10; 1; 11), providing a reliable and provable robustness guarantee to adversarial examples within specific norm bounds. In image classification, the state-of-the-art (SoTA) certified robustness to $l_2$-norm is randomized smoothing (1), which converts a deterministic base classifier into a probabilistic classifier by adding isotropic Gaussian noise to the input and returning the majority votes over the multiple predictions of noised inputs.

However, the empirical accuracy and certified bound of randomized smoothed model is not ideal because the base model, initially trained on the original data distribution, fails to capture the noise-augmented data distribution, thus unable to correctly predict the label of the original input when subjected to the corresponding Gaussian-noised counterpart (12). To address this, the base models either need to be trained from scratch to better learn the noise-augmented data distribution or incorporating a custom-trained denoiser to eliminate the Gaussian-noised inputs before they reach the base classifier (2; 50). Although these two approaches are intuitive, each of them holds its own limitation. On one hand, training from scratch is impractical for large scale models and the trade-off between time and computational cost in achieving a robust version of a large model is unworthy. On the other hand, applying a denoiser brings both training and inference time adds-on, especially when applying the SoTA diffusion-based denoiser architecture (50). In addition the certified performance of denoised smoothing largely relies on the performance of the denoising procedure.

The limitations of existing methods highlight the need for a more efficient and effective approach to further optimize the empirical performance and the robustness bound of randomized smoothing. In this work, we explore an alternative approach, changing from a reactive approach (denoised smoothing) to a proactive approach that acquires the model's ability to **learn the underlying noise-augmented data distribution**. Instead of training the model from scratch, we propose *PEFTSmoothing* to incorporate the Parameter-Efficient Fine-Tuning (PEFT) methods (64) into randomized smoothing procedure. The original goal of PEFTs is to adapt pre-trained models to downstream tasks with fewer fine-tuning parameters and lower memory usage, while maintaining

Figure 1: Illustration of the training and inference of *PEFTSmoothing* procedure on clean and adversarial images, including the Gaussian data augmentation and aggregation prediction.

performance comparable to training from scratch (64). Inspired by this, we propose *PEFTSmoothing* that achieves certified accuracy by teaching base models to learn the underlying noise-augmented data distribution with PEFT methods in both white-box and black-box manner. The intuition behind this is that the nature of PEFTs aligns with the inherent ability of large vision models, such as Vision Transformer (ViT) (17), to understand and adapt to noised data patterns, which is more efficient and effective compared to eliminating the Gaussian noise.

Another advantage of *PEFTSmoothing* is its potential to be integrated into the fine-tuning process for downstream dataset adaptations, effectively achieving dual objectives simultaneously. PEFT methods are widely adopted in the fine-tuning of large vision models to adapt them to specific downstream tasks and datasets. By combining *PEFTSmoothing* with this fine-tuning procedure, we can simultaneously obtain a robust version of the large vision model and its adaptation tailored to downstream datasets. This integrated approach significantly reduces the computational overhead typically associated with conducting separate robustness and adaptation procedures.

Figure 1 illustrates the training and inference workflow, indicated as yellow and green arrows respectively. major steps includes the Gaussian data augmentation, fine-tuning with three most widely-applied PEFT methods: prompt-tuning (45), LoRA (46), adapter (43), as well as partial fine-tuning, and the majority votes over multiple predictions. Our contributions can be summarized as follows:

- We reveal the insight that PEFT methods can successfully guide large-scale models to capture the noise-augmented data distribution with modest computational and time costs. This insight explains the success of *PEFTSmoothing* in converting a base model into a certifiably robust classifier (Section 3).
- We present *PEFTSmoothing*, a certifiable method to convert large base models, such as ViT, into robust versions. We also explore the potential of achieving certified robustness and downstream task adaption with single fine-tuning procedure (Section 4).
- We further propose black-box *PEFTSmoothing* to address scenarios where the base model cannot undergo white-box fine-tuning.
- We conduct extensive experiments on SoTA large vision model and the results demonstrate the effectiveness and efficiency of *PEFTSmoothing*. In terms of accuracy, it significantly enhances the SoTA certified accuracy on CIFAR-10. On ImageNet, *PEFTSmoothing* achieves comparable performance with a diffusion-based denoiser (Section 5).

## 2 PRELIMINARIES

The SoTA certified robustness in $l_2$-norm is randomized smoothing. In this section, we first briefly review the certified guarantee of randomized smoothing. Then, we explain denoised smoothing, the practical approach to optimize the empirical performance of randomized smoothing.

**Randomized smoothing.** Randomized smoothing (1) converts the base classifier $\mathcal{F}$ into a smoothed classifier $\mathcal{G}$ by generating the aggregated prediction over the Gaussian noise-augmented data via majority voting. Specifically, for input $x$, $\mathcal{G}$ returns the class that is most likely to be returned by the base classifier $\mathcal{F}$ under Gaussian perturbations of $x$, which can be stated as:

$$\mathcal{G}(x) = \arg\max_{c \in \mathcal{Y}} \mathbb{P}[\mathcal{F}(x + \varepsilon) = c], \text{ where } \varepsilon \sim \mathcal{N}\left(0, \sigma^2 I\right) \tag{1}$$

Under different noise scales, randomized smoothing provides a tight $l_2$ certification bound $\mathcal{R}$. Formally, the theorem can be stated as:

**Theorem 2.1.** *Given a deterministic classifier $\mathcal{F}$ and its probabilistic counterpart $\mathcal{G}$ defined in Equation 1, let $\varepsilon \sim \mathcal{N}(0, \sigma^2 I)$, suppose $c_A$ is the most probable class, and $\underline{p_A}, \overline{p_B} \in [0, 1]$ satisfy:*

$$\mathbb{P}(\mathcal{F}(x + \varepsilon) = c_A) \geq \underline{p_A} \geq \overline{p_B} \geq \max_{c \neq c_A} \mathbb{P}(\mathcal{F}(x + \varepsilon) = c) \tag{2}$$

*Then $\mathcal{G}(x + \delta) = c_A$ for all $\|\delta\|_2 < R$, where $\mathcal{R} = \frac{\sigma}{2}(\Phi^{-1}(\underline{p_A}) - \Phi^{-1}(\overline{p_B}))$*

$\underline{p_A}$ and $\overline{p_B}$ are the lower bound and upper bound of the top two possible classifications, and $\Phi^{-1}$ denotes the inverse of the standard Gaussian CDF. The intuition is to search for the radius that the lower bound of the highest class is still higher than the upper bound of the second highest class under certain Gaussian perturbation. Any adversarial examples within the $l_2$ ball with clean input $x$ as the center and $\mathcal{R}$ as the radius, are statistically proved to hold the same prediction results as $x$.

**Denoised Smoothing.** The importance of training the model from scratch with Gaussian augmented inputs has been emphasized in both randomized smoothing (1) and PixelDP (10). To overcome the computation bottleneck of retraining the large-scale models, denoised smoothing is proposed to eliminate the noise by presenting a custom-trained denoiser $\mathcal{D}_\theta$ to image classifier $\mathcal{F}$. The smoothed classifier $\mathcal{G}$ can be formulated as:

$$\mathcal{G}(x) = \arg\max_{c \in \mathcal{Y}} \mathbb{P}\left[\mathcal{F}\left(\mathcal{D}_\theta(x + \varepsilon)\right) = c\right], \text{ where } \varepsilon \sim \mathcal{N}\left(0, \sigma^2 I\right). \tag{3}$$

One major limitation of this method is that its implementation involves the training of multiple denoisers for various noise types and scales. In addition, it diminishes the model's accuracy and its performance is not satisfactory. To further improve the denoising performance, researchers later involved SoTA diffusion-based denoiser to optimize the performance of randomized smoothing (62; 50). Stable diffusion (16) are well-equipped for Gaussian denoising due to their training procedures, enabling them to effectively eliminate Gaussian noise and reconstruct clean images from noisy inputs. Although diffusion-based denoiser has largely improved the empirical accuracy of randomized smoothing, one limitation is its significantly long inference time, as each input requires multiple samplings of Gaussian augmentation which leads to multiple times passing diffusion-based denoiser.

## 3 INTUITION: PEFT GUIDES LARGE VISION MODELS TO LEARN NOISE-AUGMENTED DATA DISTRIBUTION

In this section, we discuss the intuition behind *PEFTSmoothing*, using Parameter-Efficient Fine-Tuning to achieve a certifiably robust classifier that large-scale models have the potential to learn diverse data patterns effectively, including the noise data distributions, without applying a heavy-structured denoiser. First, we will explain why randomized smoothing empirically fails without training from scratch. Second, we will demonstrate that PEFT methods can more successfully guide large vision models to learn noise-augmented data distribution than eliminate the noise with denoisers.

Theoretically, Theorem 2.1 of randomized smoothing holds regardless of how the base classifier is trained. However, if the model is not trained from scratch with Gaussian augmented data, it fails to predict accurately in the inference time with Gaussian noised inputs, thus, the defense capacity to adversarial examples as well as the certified bound are significantly diminished. In other words, the accuracy largely relies on how the model classifies the Gaussian noise-augmented data. Therefore, how to improve the model's prediction ability on noise-augmented data is the key component.

Denoised smoothing involves an auxiliary custom-trained denoiser to reactively eliminate the Gaussian noise before feeding the inputs to the certified classifier. The denoiser allows the model to predict accurately on the denoised input, which is close to the original data distribution. However, we assume that large-scale models such as Vision Transformer (ViT), acquire such potential to capture complex patterns and information if the models are trained with proper methods and training data, making them well-suited for learning intricate data distributions, including noise-augmented data which corresponds with our results in section 5.

The intuitive nature of PEFT aligns with the inherent ability of large models to understand and adapt to diverse data patterns. Inspired by this, we propose *PEFTSmoothing*, utilizing PEFT methods to guide the model to adjust its parameters more efficiently and effectively to the noise-augmented data distribution compared to the potentially sub-optimal process of training an additional denoising module. To the best of our knowledge, this is the first attempt to explore the PEFT methods in learning noised data distribution.

To substantiate our hypothesis that PEFT can more effectively guide large-scale models such as ViT to learn the noise-augmented data distribution, we compare the prediction accuracy of prompt-tuning against a DnCNN-based denoiser on noise-augmented inputs with varying Gaussian noise scales. We fine-tuned ViT-large, ViT-base, and ResNet with 0.25 Gaussian-noised data. Consequently, these models inherently achieve their highest accuracy around the 0.25 noise level due to the fine-tuning process. When evaluating these smoothed classifiers with varying levels of Gaussian noise, the trends reflect the models' ability to generalize to different noise levels.

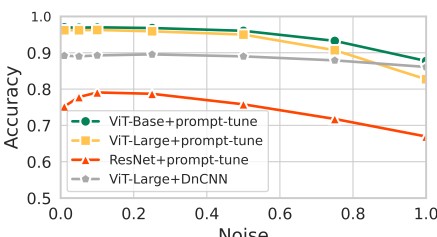

Figure 2: Comparing the noise-augmented data learning capacity of different methods. All the models are first fine-tuned with 0.25 Gaussian-noised data, and then tested the accuracy under different noise scales.

As shown in Figure 2, for both ViT-Large and ViT-Base, prompt-tuning with noise-augmented data (yellow and green lines) consistently outperforms ViT with an auxiliary denoiser (grey line) across different noise scales. These results indicate that for large-scale models, PEFT can better guide the model to learn the noise-augmented data distribution. Furthermore, upon comparing the accuracy trends of prompt-tuned ResNet (represented by the red line) with those of ViT (indicated by the yellow and green lines), it becomes evident that as model complexity increases, prompt-tuning excels in guiding the model to effectively learn the noise-augmented data distribution, resulting in higher accuracy. However, for models with the same structure but different scales (ViT-B and ViT-L), there is no significant difference in the ability to learn the Gaussian noised data.

## 4   *PEFTSmoothing*

As demonstrated in the previous section that PEFT methods have great potential to guide large vision models to learn noise-augmented data distribution, we introduce a novel paradigm *PEFTSmoothing* for robustness enhancement, which not only empirically works but still holds the certifiable guarantee of randomized smoothing as well.

In this section, we first describe the white-box *PEFTSmoothing* with prompt-tuning (45), LoRA (46) and adapter (43; 42). Second, we introduce the black-box *PEFTSmoothing*, considering the more general cases where the user wants to have a robust version of a larger model without access to the private classification APIs. At the end, we demonstrate the potential of killing two birds with one stone: achieving certified robustness and downstream task adaption with a single fine-tuning procedure.

### 4.1   WHITE-BOX *PEFTSmoothing*

The training stage and inference are illustrated in Figure 1. In the training stage, training images are augmented with random Gaussian noise, before feeding into the classifier. We incorporate four PEFT methods to transform a base classifier into a PEFTSmoothed classifier including prompt-tuning, LoRA and adapter, as well as the full fine-tuning. For all four fine-tuning methods, the blue blocks indicated the frozen layers which will not be optimized or updated during training while the light red blocks refer to the part that will be updated.

Given a base-classifier $\mathcal{F}_\theta$, a dataset with image $x$ and its corresponding correct label $y$, we follow the standard fine-tuning method to optimize the prediction of the Gaussian perturbed input $x + \varepsilon$ over ground-truth label $y$. The optimization process can be stated as follows:

$$\underset{\mathcal{F}_\theta^*}{\arg\min} \quad \mathcal{CE}Loss(y, \mathcal{F}_\theta(x + \varepsilon)), \text{where } \varepsilon \sim \mathcal{N}\left(0, \sigma^2 I\right) \tag{4}$$

where $\mathcal{CE}Loss$ refers to the standard cross entropy loss and $\mathcal{F}_\theta(\cdot)$ returns the probability vectors over the labels.

In the inference stage, images (both clean and adversarial) are also augmented with random Gaussian noise. As the PEFTSmoothed classifier $\mathcal{F}_\theta^*$ is capable of understanding and adapting to Gaussian noised patterns, we take the majority votes over the noise-augmented inputs to be the final output. This inference procedure not only give a certifiable bound around the original input but also achieves higher empirical accuracy.

### 4.2   BLACK-BOX *PEFTSmoothing*

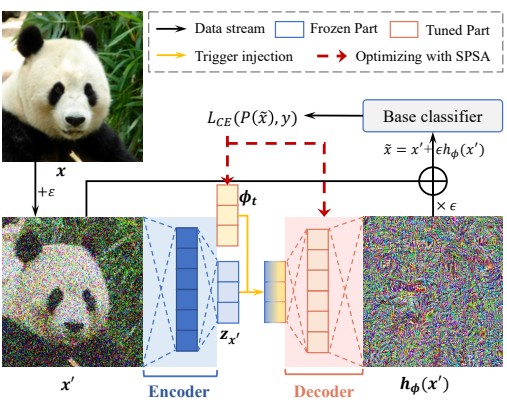

Figure 3: Illustration of black-box *PEFTSmoothing* utilizing black-box prompt-tuning.

To adapt *PEFTSmoothing* to more general cases where the base model is a black-box one that common white-box PEFT methods are not applicable, we also propose black-box *PEFTSmoothing* utilizing black-box prompt-tuning (65; 66). The advantage of adapting prompt-tuning to the black-box version is that black-box prompt-tuning is independent of the base model parameters and does not involve the modification of the main model architecture. The basic idea of black-box prompt-tuning is to generate custom pixel-style prompts by a learnable autoencoder (71) via approximating the high-dimensional gradients with Simultaneous Perturbation Stochastic Approximation (SPSA) (67; 68) which estimates the gradient of the target black-box model based on the output difference of perturbed parameters instead of directly calculating the gradients in the white-box setting. More specifically, we first build an autoencoder-based Coordinator, consisting of a frozen encoder $f(\cdot)$ and a lightweight learnable decoder $g_{\phi_d}(\cdot)$ with parameter $\phi_d$. The pixel-style prompts are generated by the Coordinator and added to the image $x$. To have a prompt-injected data with the Gaussian noise, we have:

$$\tilde{x} = clip(x' + \epsilon h_\phi(x')), \text{where} \quad h_\phi(x') = g_{\phi_d}(z_{x'}, \phi_t), \text{and} \quad x' = x + \varepsilon \tag{5}$$

where $\varepsilon \sim \mathcal{N}\left(0, \sigma^2 I\right)$, $x'$ is the Gaussian noised image and $\phi_t$ is a task-specific prompt trigger vector that is jointly optimized with decoder parameter $\phi_d$. Here $z_{x'} = f(x')$ is the output of the encoder and $\epsilon \in [0, 1]$ is the hyperparameter to control the power of the prompt. The procedure of black-box *PEFTSmoothing* is demonstrated in Figure 3.

### 4.3 TWO BIRD ONE STONE

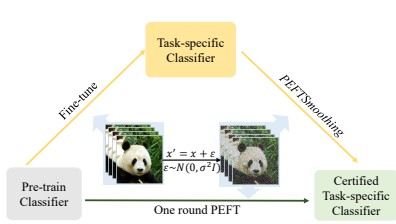

Figure 4: One round of PEFT to achieve both *PEFTSmoothing* and downstream dataset adaptation

Another noteworthy benefit of *PEFTSmoothing* is its capability to integrate into the fine-tuning process for adapting large vision models to various downstream datasets and tasks. This integration allows us to kill two birds with one stone: enhancing the robustness of the model and customizing its adaptation to specific datasets. PEFT methods are commonly employed in fine-tuning large vision models to meet the demands of specific downstream applications. By incorporating *PEFTSmoothing* into this fine-tuning workflow, we can efficiently obtain a robust and tailored version of the vision model, optimized for downstream datasets. As shown in Figure 4, one round of PEFT using the Gaussian augmented dataset can serve as a shortcut to obtaining a certified task-specific classifier. This combined approach significantly minimizes the computational costs that would otherwise be incurred by conducting separate robustness enhancement and adaptation procedures.

## 5 EXPERIMENTS

In this section, we first elaborate on the experiment configurations in section 5.1. Second, we evaluate *PEFTSmoothing* on the Vision Transformer (ViT) for CIFAR-10 and ImageNet in terms of certified accuracy and computation costs, compared with baseline methods including basic randomized smoothing and denoised smoothing in section 5.2. Next, we use Gradient-weighted Class Activation Mapping (Grad-CAM) (13) to explore the reason for the different performance of *PEFTSmoothing* on CIFAR-10 and ImageNet in section 5.3. Following that, we conduct ablation studies on prompt-tuning and LoRA to demonstrate how the selection of hyper-parameters influences the certified results in section 5.5. In addition, we further evaluate the black-box *PEFTSmoothing* from the perspective of certified accuracy in section 5.6. Last, we present the result of killing two birds with one stone, showing the possibility of integrating the fine-tuning of *PEFTSmoothing* with PEFT methods intended for downstream dataset adaptation in section 5.7.

| Category | Method | CIFAR-10 | | | |
|---|---|---|---|---|---|
| | | $\sigma = 0.25$ | $\sigma = 0.50$ | $\sigma = 0.75$ | $\sigma = 1.00$ |
| | PixelDP (10) | $22.0^{(71.0)}$ | $2.0^{(44.0)}$ | - | - |
| | RS (1) | $61.0^{(75.0)}$ | $43.0^{(75.0)}$ | $32.0^{(65.0)}$ | $22.0^{(66.0)}$ |
| | SmoothAdv (54) | $67.4^{(75.6)}$ | $57.6^{(75.6)}$ | $47.8^{(74.7)}$ | $38.3^{(57.4)}$ |
| | SmoothAdv (54) | $74.9^{(74.3)}$ | $63.4^{(80.1)}$ | $51.9^{80.1}$ | $39.6^{(62.2)}$ |
| RS | Consistency (56) | $68.8^{(77.8)}$ | $58.1^{(75.8)}$ | $48.5^{72.9}$ | $37.8^{(52.3)}$ |
| | MACER (63) | $71.0^{(81.0)}$ | $59.0^{(81.0)}$ | $46.0^{(66.0)}$ | $38.0^{(66.0)}$ |
| | Boosting (57) | $70.6^{(83.4)}$ | $60.4^{(76.8)}$ | $52.4^{71.6}$ | $38.8^{(52.4)}$ |
| | DRT (58) | $70.4^{(81.5)}$ | $60.2^{(72.6)}$ | $50.5^{71.9}$ | $39.8^{(56.1)}$ |
| | SmoothMix (59) | $67.9^{(77.1)}$ | $60.4^{(76.8)}$ | $52.4^{71.6}$ | $38.8^{(52.4)}$ |
| | ACES (60) | $69.0^{(79.0)}$ | $57.2^{(74.2)}$ | $47.0^{74.2}$ | $37.8^{(58.6)}$ |
| | Denoised (2) | $56.0^{(72.0)}$ | $41.0^{(62.0)}$ | $28.0^{(62.0)}$ | $19.0^{(44.0)}$ |
| DS | Lee (62) | $60.0$ | $42.0$ | $28.0$ | $19.0$ |
| | Diffusion (50) | $76.7^{(88.1)}$ | $63.0^{(88.1)}$ | $45.3^{(88.1)}$ | $32.1^{(77.0)}$ |
| | LoRA(Ours) | **97.8**±9.3E-6$^{(97.0)}$ | **97.5**±5.3E-5$^{(95.4)}$ | **95.8**±5.2E-5$^{(94.4)}$ | **94.8**±1.2E-5$^{(91.4)}$ |
| PEFTSmoothing | Prompt-tune(Ours) | 96.0±3.7E-5$^{\mathbf{(97.2)}}$ | 93.6±2.0E-6$^{\mathbf{(97.2)}}$ | 92.5±4.8E-5$^{\mathbf{(95.8)}}$ | 85.6±4.8E-5$^{\mathbf{(94.0)}}$ |
| | Full fine-tune(Ours) | 92.7±6.4E-5$^{(94.0)}$ | 86.9±2.3E-5$^{(89.0)}$ | 85.8±4.8E-5$^{(84.8)}$ | 82.9±4.8E-5$^{(82.8)}$ |
| | Adapter(Ours) | 93.7±1.0E-6$^{(91.2)}$ | 91.1±2.5E-4$^{(89.8)}$ | 86.5±1.5E-5$^{(90.2)}$ | 82.1±4.8E-5$^{(87.4)}$ |

Table 1: Certified accuracy of randomized smoothing, denoised smoothing and their variants, and *PEFTSmoothing*. We report in the form of mean ± variance. Each entry lists the certified accuracy, with the clean accuracy for that model in parentheses, using numbers taken from respective papers to demonstrate the certainty of *PEFTSmoothing*.

## 5.1 CONFIGURATION

We evaluate *PEFTSmoothing* results on two standard datasets, CIFAR-10 (14), consisting of 60000 32x32 color images in 10 classes, and ImageNet (47) which is a 1000-classification task. All experiments of ImageNet are conducted on a single A100 GPU and CIFAR-10 on a single A40 GPU.

**Model configuration.** The base classifier we used to test performance of *PEFTSmoothing* on CIFAR-10 is a 86.6M-parameter ViT-B/16 model (17) pre-trained on ImageNet-21k (47) and fine-tuned to CIFAR-10. For ImageNet, we also used the same pre-trained ViT-B/16 model but fine-tuned on ImageNet2012 by Google (17).

**Baselines.** For baseline comparison, we mainly compared the performance and the computation costs of *PEFTSmoothing* with the SoTA denoised smoothing, namely DnCNN-based (48) and diffusion-based denoiser (49). Besides, we also use a state of art SUNet-structured (swin transformer + UNet) denoiser (53) which is used in medical image denoising as an additional baseline denoiser.

**PEFTSmoothing configuration.** We evaluate *PEFTSmoothing* with the three most popular PEFT methods, namely prompt-tuning, LoAR, and adapter. For prompt-tuning, we add soft prompts as prefixes before the input embedding of an image with a length of 100. For LoRA, we add trainable rank decomposition matrices into each layer of the Transformer architecture with a rank of 2. For the adapter, we insert new MLP modules with residual connections inside Transformer layers. The hyper-parameter selection and ablation study will be discussed in the sections below. For the black-box *PEFTSmoothing*, we adopt a CLIP ViT-B/16 (4) as the pre-trained model and an ImageNet pre-trained *vit-mae-base* as the frozen encoder of Coordinator which is introduced above in Section 4.

**Evaluation metrics.** Certified accuracy is the standard metric to evaluate the robustness of the defense methods. Certified accuracy denotes the fraction of the clean testing set on which the predictions are both correct and satisfy the certification criteria (see Theorem 2.1). Formally, it is defined as:

$$CA = \frac{\sum_{t=1}^{T} certifiedCheck(x, \varepsilon) \& corrClass(x, \varepsilon)}{T}, \tag{6}$$

where $certifiedCheck$ returns 1 if Theorem 2.1 is satisfied and $corrClass$ returns 1 if the classification output is correct.

Besides, we also use the size of parameters involved in the training process to reflect the training computation cost. As the certified robustness methods involves the aggregation prediction of Gaussian noise inputs, we also demonstrate the average inference time for the prediction of a single input to reflect the latency in the inference stage.

## 5.2 CERTIFIED ACCURACY OF *PEFTSmoothing*

In Figure 5, we demonstrate the certified accuracy of *PEFTSmoothing* with each PEFT method (orange, black and blue lines), as well as the comparison with the baseline methods including different based denoised smoothing (dark green, red and purple line), full fine-tuning with Gaussian augmentation (grey line) and base model without training on Gaussian augmentation (light green line). The first two figures and the latter two figures represent the certified accuracy on CIFAR-10 and ImageNet respectively under different Gaussian noise scales $\sigma$. Note that for denoised smoothing, we only demonstrate the best denoiser results, DnCNN-based and SUNet-based on CIFAR-10 and Diffusion-based denoiser on ImageNet.

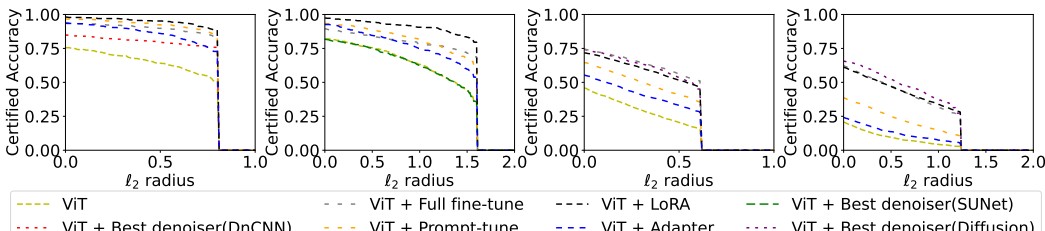

Figure 5: Certified accuracy on *PEFTSmoothing*, denoised smoothing, and randomized smoothing without training on Gaussian noise data. Results are conducted on CIFAR-10 and ImageNet, with different Gaussian noise scale, $= 0.25$ and $\sigma = 0.5$. **First**: CIFAR-10, $\sigma = 0.25$, **Second**: CIFAR-10, $\sigma = 0.50$, **Third**: ImageNet, $\sigma = 0.25$, **Forth**: ImageNet, $\sigma = 0.50$

As shown in first two figures of Figure 5, *PEFTSmoothing* outperforms all the SoTA denoised smoothing approaches on CIFAR-10. *PEFTSmoothing* has not only the highest certified accuracy when $l_2$ radius = 0, but also decreases the most gradually when $l_2$ radius increases. Surprisingly, LoRA and Prompt-tune even outperform full fine-tuning on both noise scales ($\sigma = 0.25$ and $\sigma = 0.5$), with training significantly fewer total model parameters than full fine-tuning (LoRA: 0.081M, Prompt-tune: 0.929M, adapter: 3.387M *vs* Full: 86.6M). This trend is also observed in other applications of PEFT methods in other fields (45). Therefore, even though storage is not a concern, *PEFTSmoothing* is a promising approach for processing a base classifier for randomized smoothing.

Nevertheless, the experiments on ImageNet (latter two figures of Figure 5) experience a different trend where the denoised smoothing with Diffusion-based denoiser has a slightly higher certified accuracy than *PEFTSmoothing* with LoRA. This is mainly due to the powerful denoising ability inherent in the intricate diffusion architecture, especially for high-resolution images. However, it's noteworthy that while this complex architecture enhances the denoising process, it simultaneously results in significantly prolonged inference times for individual examples, which is visualized in Table 2, given that all noise-augmented data must traverse the Diffusion-based denoiser.

To conduct a more thorough comparison with basic randomized smoothing methods as well as the denoised smoothing, in Table 1 and Table 4 in Appendix A, we report the top-1 certified accuracy achieved by *PEFTSmoothing* and other baseline methods for different noise magnitudes on two datasets respectively. For results of *PEFTSmoothing* in CIFAR-10, we conduct all the experiments for three times and report in the form of mean $\pm$ variance.

Table 1 indicates the results on CIFAR-10, where *PEFTSmoothing* outperforms all baseline randomized smoothing and denoised smoothing methods at all noise magnitudes greatly. All four versions of *PEFTSmoothing* methods can achieve over 80% top-1 certified accuracy at high $\sigma$ distortions ($\sigma >= 0.5$) and can achieve over 90% at low $\sigma$ distortions while the state-of-the-art denoised smoothing and randomized smoothing methods can achieve at most around 75% at $\sigma = 0.25$. Furthermore, among all four *PEFTSmoothing* methods, LoRA performs the best top-1 accuracy which can maintain over 94% over all $\sigma$ and prompt-tune performs the best accuracy on clean dataset but top-1 certified accuracy is slightly worse than LoRA.

### 5.3 GRAD-CAM ANALYSIS

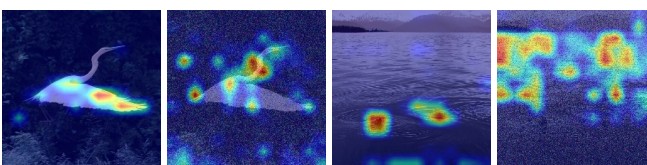

(a) bird(clean)    (b) bird(noised)    (c) fish(clean)    (d) fish(noised)

Figure 6: Different clean and Gaussian noised images predicted by normal classifier and *PEFTSmoothed* classifier

As is shown in the third and forth figure of Figure 5, LoRA achieves slightly lower performance than diffusion based denoised smoothing. We further discover that the reason behind this phenomenon is that *PEFTSmoothed* classifier can classify correctly for most examples in ImageNet, but cannot achieve a high logit probability for the top-1 class, failing to pass the confidence test according to equation 2 with low $\underline{p_A}$ of *PEFTSmoothed* classifiers.

To further support our findings, in this section, we use Gradient-weighted Class Activation Mapping (Grad-CAM) (13) to explore the reason for the different improvements of certified accuracy achieved by *PEFTSmoothing* on CIFAR-10 and ImageNet. Grad-CAM is a technique that helps visualize

which parts of an image are most important for prediction. The visualization of our findings is demonstrated in Figure 6, which highlights the regions of an input image that are most important for the network's prediction by producing a heatmap overlaid on the image. The original image of Figure 6 is a crane flying in the forest and the original image of Figure 6c is fish in the lake. Figure 6a and 6c are the Grad-CAM results of the clean images predicted by a base ImageNet classifier and Figure 6b and 6d are the Grad-CAM results of the Gaussian noised images predicted by a LoRA based *PEFTSmoothed* classifier. As is shown in Figure 6a and 6c, most highlighted regions are located on the main features of the pattern in the images. However, for the Gaussian noised images, the highlighted regions become random and unordered as it is all over the whole image, which explains the low confidence for the top-1 class. As a result, we believe that diffusion-based denoisers have more powerful performance on high-resolution images with large $\sigma$ Gaussian noise as it has high denoising ability which eliminates the influence of the noise on the image. The detailed results on ImageNet is included in Appendix A.

## 5.4 COMPUTATION COST OF *PEFTSmoothing*

| Method | Denoised Smoothing | | | *PEFTSmoothing* | | | |
|---|---|---|---|---|---|---|---|
| | SUNet | Diffusion | DnCNN | Full Fine-Tune | Adaptor | LoRA | Prompt-Tune |
| Parameters | 99.7M | 52.5M | 0.558M | 86.6M | 3.39M | 0.081M | 0.929M |
| Inference Time on CIFAR-10 | 16s | 37s | 119s | 18s | 18s | 10s | 29s |
| Inference Time on ImageNet | - | 120s | 96s | 1.94s | 4.51s | 1.89s | 3.14s |

Table 2: Comparison of the computational cost from the perspective of trained parameters and inference time of certifying CIFAR-10 and ImageNet. Base classifier is ViT-base.

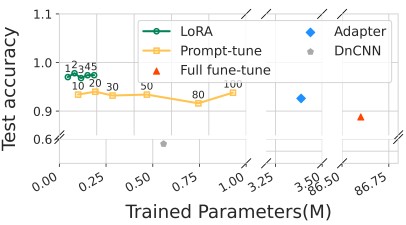

Figure 7: Test Accuracy vs. Size of Trained Parameters

In terms of efficiency, we demonstrate the size of training parameters and inference time in Table 2. *PEFTSmoothing* with LoRA, adapter, and prompt-tuning reduces the training parameters by 1000 times compared to Diffusion-based denoisers and 10 times compared to DnCNN-based denoisers. This significant reduction in training parameters indicates substantial savings in computational costs for obtaining a certifiably robust classifier. In addition, we compare the latency in the inference stage of different methods by certifying CIFAR-10 and ImageNet images on a single A40 GPU, setting the noise sampling number to N=10000 on CIFAR-10 and N=1000 on ImageNet, indicating the savings of *PEFTSmoothing* in terms of the inference time.

To give a more intuitive sense, We also demonstrate the trade-off between the size of trained parameters and achieved certified accuracy in Figure 7, comparing *PEFTSmoothing* with the denoised smoothing under different ablation settings. Ideally, we want to achieve high certified accuracy with a low size of trained parameters (upper left part of the graph), which is dominated by *PEFTSmoothing* with LoRA and prompt-tuning. In terms of inference time, *PEFTSmoothing* surpasses the SoTA $l_2$-norm certified defense.

## 5.5 CIFAR-10 ABLATION STUDIES

| LoRA Rank | Radius | | | | | Prompt length | Radius | | | |
|---|---|---|---|---|---|---|---|---|---|---|
| | R=0 | R=0.5 | R=1 | R=1.5 | | | R=0 | R=0.5 | R=1 | R=1.5 |
| Rank=1 | 0.97 | 0.94 | 0.878 | 0.778 | | Length=10 | 0.938 | 0.914 | **0.874** | 0.772 |
| Rank=2 | **0.978** | 0.94 | 0.894 | 0.806 | | Length=20 | **0.94** | **0.916** | 0.852 | **0.764** |
| Rank=3 | 0.968 | **0.954** | **0.908** | **0.848** | | Length=30 | 0.932 | 0.912 | 0.846 | 0.756 |
| Rank=4 | 0.974 | 0.944 | 0.906 | 0.832 | | Length=50 | 0.934 | 0.888 | 0.836 | 0.74 |
| Rank=5 | 0.974 | 0.944 | 0.906 | 0.848 | | Length=80 | 0.916 | 0.886 | 0.828 | 0.742 |
| | | | | | | Length=100 | 0.934 | 0.888 | 0.816 | 0.676 |

Table 3: Ablation study on *PEFTSmoothing*. **Left**: LoRA ranks. **Right**: prompt length.

In this section, we present the ablation studies of *PEFTSmoothing* on CIFAR-10 with prompt-tuning and LoRA in terms of prompt lengths and LoRA ranks respectively, aiming to investigate the impact of varying hyper-parameters on the performance of the *PEFTSmoothing*. For prompt-tuning, the fine-tuning performance is heavily rely on the selection of pre-defined prompt length. Left figure of Figure 8 shows the certified accuracy of *PEFTSmoothing* with prompt-tuning under different prompt lengths when setting noise scale $\sigma$ to 0.5. As illustrated in the figure, under the same $l_2$ radius, smaller prompt lengths can achieve higher certified accuracy. To further support this finding, we present the certified accuracy under different radii with different prompt lengths in left of Table 3, where prompt length equal to 20 can generally achieve the highest certified accuracy.

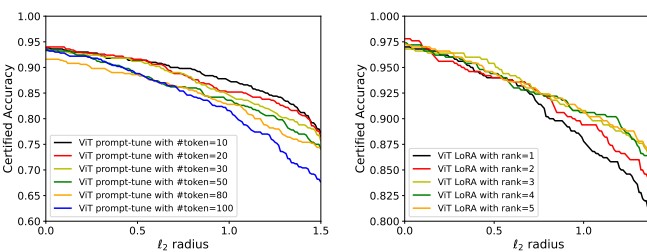

Similarly, we study the impact of the LoRA rank over the *PEFTSmoothing* certified accuracy in right figure of Figure 8 and right of Table 3, where rank equals 3 demonstrates the highest certified accuracy.

Figure 8: Ablation study on *PEFTSmoothing*. **Left**: Certified Accuracy of prompt-tuning in *PEFTSmoothing* with Different Prompt Lengths. $\sigma = 0.5$. **Right**: Certified Accuracy of prompt-tuning with Different Prompt Lengths. $\sigma = 0.5$

Notably, even with as few as only 10 prompts or $rank = 1$, *PEFTSmoothing* still achieves high certified accuracy and remains competitive or even better compared to full fine-tuning and other certified defense methods, indicating the ability to guide the model to learn the noised inputs. In addition, we also found that different certified radii may require different hyperparameters to achieve optimal certified accuracy. Table 3 reveal a bell-shaped relationship between the rank and the results, enabling us to make an optimal selection of rank and prompt length under different radii.

## 5.6 BLACK-BOX *PEFTSmoothing*

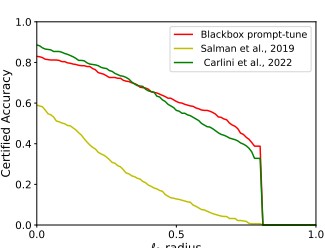

To demonstrate the effectiveness of black-box *PEFTSmoothing*, we present a comparative analysis of its certified accuracy on the CIFAR-10 dataset against two certified defense methods under black-box settings: DnCNN-based denoiser (2) and Diffusion-based denoiser (50) in Figure 9. It is important to note that while the DnCNN-based denoiser does not require fine-tuning the base model directly, it still requires access to the base model's gradients during the fine-tuning process of the denoiser itself. This grey-box setup differs from the true black-box nature of PEFTSmoothing.

Figure 9: Certified Accuracy of Black-box *PEFTSmoothing*

As illustrated in the figure, black-box *PEFTSmoothing* greatly outperforms DnCNN-based denoised smoothing (2) with top-1 certified accuracy of 0.83 against 0.6. Meanwhile, black-box *PEFTSmoothing* can achieve similar performance to diffusion-based denoised smoothing (50) since our method has better-certified accuracy at large radius ($l_2 > 0.4$) and nearly match it at small radius.

## 5.7 TWO BIRDS ONE STONE

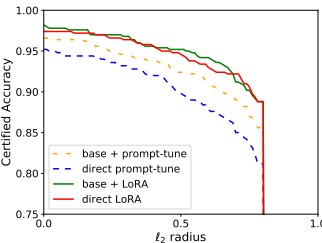

In this section, we present the possibility of integrating the fine-tuning of *PEFTSmoothing* with PEFT intended for downstream dataset adaptation. Specifically, we choose a ViT-B/16 model pre-trained on ImageNet-21k as the base model and CIFAR-10 as the downstream dataset with *PEFTSmoothing* noise scale $\sigma = 0.25$. For both training methods, we train the model for 50 epochs in total. In full line of Figure 10 we compare one round of prompt-tuning to achieve both *PEFTSmoothing* and downstream dataset (blue lines) with fine-tuning to the datasets and *PEFTSmoothing* sequentially (yellow lines). The same experiment is conducted on LoRA in dotted line of Figure 10. Results demonstrate that these two strategies can achieve comparable performance of certified accuracy, especially with LoRA.

Figure 10: Integrating the fine-tuning of different based *PEFTSmoothing* with PEFT intended for downstream dataset adaptation.

Considering the prevailing practice in image classification, where models are commonly fine-tuned from pre-trained ones to adapt to the specific downstream datasets, these results indicate the promising direction of achieving a certifiable robust version of deep learning systems together with downstream adaptations for free.

Generally, the experimental findings demonstrate that PEFTSmoothing surpasses existing certified defense methods on CIFAR-10 and ImageNet, while also reducing the computational overhead of the defense significantly. Additionally, we investigated the applicability of black-box PEFTSmoothing and the feasibility of adapting a PEFTSmoothed model to downstream datasets through fine-tuning.

## 6 RELATED WORK

### 6.1 ADVERSARIAL EXAMPLES

**Adversarial Attacks.** The concept of adversarial attacks was initially introduced by Szegedy et al. (36), who highlighted the susceptibility of neural networks to small perturbations in input data, capable of deceiving machine learning models. This foundational work brought significant attention to the field of adversarial machine learning. Following this, Goodfellow et al. proposed the Fast Gradient Sign Method (FGSM), which efficiently generates adversarial examples by using the gradient of the loss function with respect to the input data (7). **Adversarial training.** Among empirical defenses, adversarial training has emerged as one of the most successful approaches. Initially proposed by Goodfellow et al. (7), adversarial training involves incorporating adversarial examples into the training dataset, thereby enhancing the model's robustness. Madry et al. (34) extended this idea by formulating adversarial training as a min-max optimization problem, which further improved the resilience of models against adversarial attacks. **Empirical defense.** Besides adversarial training, there are some other attack strategies adversarial detection (61; 27; 28; 51; 52) and gradient masking. However, Carlini and Wagner (23) summarized most of these adversarial detecting methods cannot defend adversarial examples in some cases by slightly changing the loss functions. In addition, many heuristic defenses were later compromised by stronger adversarial attack methods, as demonstrated by Athalye et al. (8) and Uesato et al. (72), who showed that many defenses relied on obfuscated gradients, which provided a false sense of security and could be easily bypassed.

### 6.2 CERTIFIED ROBUSTNESS

Certified defenses aim to provide provable guarantees on the robustness and reliability of models against adversarial attacks. Lecuyer et al. (10) introduced PixelDP, which scales to large networks and datasets by establishing a connection between robustness against adversarial examples and differential privacy. This method provides formal guarantees on the model's robustness by leveraging the principles of differential privacy (55). Building on this concept, Cohen et al. (1) developed randomized smoothing, a technique that creates robustness guarantees by adding random noise to the input data and averaging the model's predictions over multiple noisy samples.

Subsequent research has focused on improving randomized smoothing to maximize its empirical performance. For instance, Salman et al. (54) integrated adversarial training into the randomized smoothing framework to further improve its defense capacity against adversarial examples. Zhai et al. (63) sought to regularize the prediction consistency over noise, thereby enhancing the robustness of the smoothed classifiers. Additionally, Kariyappa and Qureshi (37) investigated the robustness of ensemble models, demonstrating that diverse ensembles could provide better robustness compared to single models. Jeong and Song (38) proposed SmoothMix, a training scheme that enhances the robustness of smoothed classifiers through self-mixup, which blends inputs to create new training examples that help the model generalize better. Horvath et al. (40) explored the trade-off between robustness and accuracy, proposing a compositional architecture that balances these two objectives.

### 6.3 MODEL FINE-TUNING

Fine-tuning enhances a pre-trained model's performance on specific tasks but can be costly for large models. To address this, researchers propose parameter-efficient fine-tuning, optimizing model parameters with minimal resource use For example, Houlsby et al. (42) introduced adapter layers, which add a small number of trainable parameters to the model, allowing for efficient fine-tuning. Pfeiffer et al. Lester et al. (44) developed prompt-tuning, which adjusts only the prompts used to guide the model's predictions, significantly reducing the number of parameters that need to be updated. Hu et al. (46) proposed LoRA (Low-Rank Adaptation), which fine-tunes the model by learning low-rank adaptations of the weight matrices, thereby reducing computational overhead.

## 7 CONCLUSION

In this paper, we present *PEFTSmoothing* to proactively adapt the base model to learn the Gaussian noise-augmented data distribution with Parameter-Efficient Fine-Tuning methods. PEFTSmoothed model can achieve high certified accuracy when applying randomized smoothing procedures. We experimented *PEFTSmoothing* with different PEFT strategies and compared them with basic randomized smoothing and denoised smoothing. Experimental results indicate that *PEFTSmoothing* greatly outperforms the existing certified defense methods on CIFAR-10 and ImageNet while significantly decreasing the computational cost of the defense. We further explored the black-box *PEFTSmoothing* and the possibility of achieving a PEFTSmoothed model along with fine-tuning to adapt to downstream datasets. Extensive experiments demonstrate the effectiveness and efficiency of *PEFTSmoothing*.

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

## A  DETAILED ANALYSIS ON RESULTS OF IMAGENET

we report the top-1 certified accuracy achieved by *PEFTSmoothing* and other baseline methods for different noise magnitudes on ImageNet in Table 4.

The results on ImageNet, reveal the same trend as third and forth figure in figure 5, that denoised smoothing with Diffusion-based model has the best certified accuracy at high $\sigma$ distortions ($\sigma > 0.5$), around 10% higher than the best *PEFTSmoothing* with LoRA. As for Gaussian augmented data with $\sigma < 0.5$, *PEFTSmoothing* achieves similar performance with the SoTA diffusion-based results. Most of the results in the table are referenced from the original paper and have been verified by reproducing similar results. However, some results for certain noise scales, specifically $\sigma = 0.1$ and $\sigma = 0.25$, are not reported in the original paper and are indicated with "-".

| CATEGORY | METHOD | ImageNet $\sigma = 0.10$ | $\sigma = 0.25$ | $\sigma = 0.50$ | $\sigma = 1.00$ |
|---|---|---|---|---|---|
| RS | PIXELDP (10) | - | - | $16.0^{(33.0)}$ | - |
| | RS (1) | - | 66.7 | $49.0^{(67.0)}$ | $37.0^{(57.0)}$ |
| | SMOOTHADV (54) | - | 63.0 | $56.0^{(65.0)}$ | $43.0^{(54.0)}$ |
| | SMOOTHADV (54) | - | 67.0 | - | - |
| | CONSISTENCY (56) | - | 64.8 | $50.0^{(55.0)}$ | $44.0^{(55.0)}$ |
| | MACER (63) | - | 68.0 | $57.0^{(68.0)}$ | $43.0^{(64.0)}$ |
| | BOOSTING (57) | - | 65.6 | $57.0^{(65.6)}$ | $44.6^{(57.0)}$ |
| | DRT (58) | - | - | $46.8^{(52.2)}$ | $44.4^{(55.2)}$ |
| | SMOOTHMIX (59) | - | - | $50.0^{(55.0)}$ | $43.0^{(55.0)}$ |
| | ACES (60) | - | 63.2 | $54.0^{(63.8)}$ | $42.2^{(57.2)}$ |
| DS | DENOISED (2) | $69.2^{(70.9)}$ | $58.0^{(59.8)}$ | $33.0^{(60.0)}$ | $14.0^{(38.0)}$ |
| | LEE (62) | - | - | 41.0 | 24.0 |
| | DIFFUSION (50) | $78.0^{(84.6)}$ | $74.3^{(80.3)}$ | $\mathbf{71.1}^{(82.8)}$ | $\mathbf{54.3}^{(77.1)}$ |
| PEFT | LORA(OURS) | $76.7^{(55.5)}$ | $71.72^{(28.6)}$ | $61.08^{(3.9)}$ | $39.0^{(1.00)}$ |
| | PROMPT-TUNE(OURS) | $72.8^{(71.0)}$ | $64.6^{(62.7)}$ | $38.72^{(53.9)}$ | $16.0^{(42.2)}$ |
| | FULL FINE-TUNE(OURS) | $77.4^{(73.0)}$ | $62.7^{(58.4)}$ | $62.36^{(44.1)}$ | $34.7^{(18.3)}$ |
| | ADAPTER(OURS) | $69.8^{(64.7)}$ | $55.44^{(53.6)}$ | $24.12^{(23.9)}$ | $11.4^{(0.080)}$ |

Table 4: ImageNet certified top-1 accuracy for prior defenses of randomized smoothing, denoised smoothing, and *PEFTSmoothing*. Each entry lists the certified accuracy, with the clean accuracy for that model in parentheses, using numbers taken from respective papers.

## B    DISCUSSION

**Limitation.** Overall, *PEFTSmoothing* achieves SoTA certified accuracy with significantly decreased computational cost in both white-box and black-box settings on CIFAR-10, while achieving slightly worse performance than diffusion-based denoised smoothing methods on ImageNet.

**Deployment.** Regarding the aforementioned limitation, we recommend the following deployment guidelines to provide a comprehensive understanding of the strengths and limitations of *PEFTSmoothing* across different scenarios:

- *PEFTSmoothing* outperforms all state-of-the-art denoising smoothing approaches on the CIFAR-10 dataset. It achieves this superior performance while requiring the training of significantly fewer total model parameters, making it a more efficient option in terms of computational resources and training time.

- When applied to high-resolution images such as those in the ImageNet dataset, *PEFTSmoothing* combined with LoRA exhibits slightly lower certified accuracy compared to the diffusion-based denoising smoothing method.

- Among the four *PEFTSmoothing* methods evaluated, LoRA and prompt-tuning stand out by achieving better top-1 accuracy with a smaller size of trained parameters, making them preferable choices for scenarios where parameter size and computational load are critical considerations.

- The black-box *PEFTSmoothing* using prompt-tuning can achieve performance levels similar to those of the diffusion-based denoising smoothing methods.

**Future work.** In future research, we aim to enhance the performance of *PEFTSmoothing* on high-resolution images exploring more effective PEFT methods for large-scale models. Additionally, we plan to further investigate the integration of fine-tuning *PEFTSmoothing* with PEFT for downstream dataset adaptation. This approach presents a pathway toward scalable fine-tuning algorithms for certified task-specific classifiers. Furthermore, it would be interesting to explore adapting *PEFTSmoothing* to the certified robustness method in the large language model domain.

