# OpenReview forum: "Certified PEFTSmoothing: Parameter-Efficient Fine-Tuning with Randomized Smoothing"
_ICLR.cc/2025/Conference — ICLR 2025 Conference Withdrawn Submission_

### Official Review · Reviewer_Q9gT · 2024-10-28

**Soundness:** 2
**Presentation:** 3
**Contribution:** 1
**Rating:** 3
**Confidence:** 4

**Summary:**

This work proposes a parameter-efficient fine-tuning (PEFT) method to explore the large-scale model's potential to learn the Gaussian noised data pattern, thus enhancing its performance on randomized smoothing. The white-box PEFT and black-box PEFT smoothing methods are proposed for different scenarios. Extensive and comprehensive experiments are conducted to demonstrate the effectiveness of the proposed PEFT method.

**Strengths:**

1. This paper is well-written and structured. The method is intuitive and easy for readers to understand.

2. Comprehensive experiments are conducted to demonstrate its effectiveness.

**Weaknesses:**

1. My major concern is related to the novelty and significance of this work:

**Novelty**: The white-box and black-box tuning methods in this paper are all existing work. This paper just utilizes the existing PEFT on a large-scale classification model and tests its performance under randomized smoothing. This raises concerns regarding the paper’s technical contribution.

**Significance**: The primary challenge in randomized smoothing (RS) is not training noise-robust classifiers but reducing the extensive inference time required for certification, often involving a majority vote over 10,000 samples. While the paper aims to reduce model training time through PEFT, this does not address the more critical issue of RS's substantial inference delay. This raises concerns about the practical impact of the proposed work.

2. Some experimental settings are unfair:

In Figure 2, it's unfair to compare the PEFT methods with those training-free denoising methods.

In Table 1, clarification is needed regarding the base models used for RS and DS methods. Based on the reported performance, it appears these methods are evaluated using ResNet architectures, while PEFTsmoothing employs ViT-L and ViT-B models.

3. An interesting result is that the LORA-based tuning method outperforms the full fine-tuning-based method. Typically, the performance of parameter-efficient methods could be close to but not surpass the full fine-tuning method. A more detailed analysis of this would be valuable.

**Questions:**

see the weakness.

**Details Of Ethics Concerns:**

No ethics concerns.

---

### Official Review · Reviewer_fmc4 · 2024-10-31

**Soundness:** 2
**Presentation:** 2
**Contribution:** 1
**Rating:** 3
**Confidence:** 3

**Summary:**

The paper proposes PEFTsmoothing, an approach for teaching the base model to learn the Gaussian noise augmented data for white box and black box certified robustness. Certification results are shown on benchmark datasets.

**Strengths:**

The authors present PEFTSmoothing, a certifiable method to convert large base models into robust versions. They also explore how it can be extended to achieve certified robustness and downstream task adaption with fine-tuning. Black-box extensions are considered. Experiments are shown on large vision models and results show the effectiveness of the approach.

**Weaknesses:**

1. Lack of novelty. The training objective (4) is not novel, see baselines below. The overall premise of the paper does not seem novel enough for ICLR standards. Combining two well known methods, smoothing and PEFT, does not cross the novelty threshold. There is also a very large literature on certified robustness that the authors do not compare against, see https://sokcertifiedrobustness.github.io/

2. Limited baselines. Compare against and cite the following relevant works:
- S. Srinivas et al., Which Models have Perceptually-Aligned Gradients? An Explanation via Off-Manifold Robustness, NeurIPS 2023
- R. Shao et al., On the Adversarial Robustness of Vision Transformers, TMLR 2022, https://arxiv.org/pdf/2103.15670
- T. Tsiligkaridis et al, Diverse Gaussian Noise Consistency Regularization for Robustness and Uncertainty Calibration, IJCNN 2023, https://arxiv.org/pdf/2104.01231

The datasets are also quite limited and simple so I suggest a more thorough experimental study involving more domain-specific datasets, e.g. WILDS.

3. How far is empirical robustness from certified robustness in your setting? A comparison is needed. Approaches for gradient masking mitigation have been studied. What are the tradeoffs? What about computational complexity?

4. How does the geometry of the loss landscape depend upon the robustness properties of the large vision models you study? These are intimately related based the adversarial robustness literature.

**Questions:**

See weaknesses.

---

### Official Review · Reviewer_5MLq · 2024-11-02

**Soundness:** 2
**Presentation:** 2
**Contribution:** 2
**Rating:** 3
**Confidence:** 3

**Summary:**

This work proposes a new approach to the random smoothing approach for making neural networks robust to adversarial perturbation. Rather than retraining models from scratch, this work proposes using parameter efficient fine-tuning (PEFT), typically used to fine-tune language models, to enable models to learn adversarially perturbed data-distributions.

The key contributions are as follows:
- White-box PEFT smoothing, in which a vision model is fine-tuned after perturbing the dataset with gaussian noise.
- Black-box PEFT tuning, in which zeroth-order optimization (specifically SPSA) is used to create a prompt-tuning based method for fine-tuning the model with perturbed data.
- A thorough experimental validation shows that PEFT can be used to efficiently make ViT models more robust.

**Strengths:**

- The fundamental idea of using fine-tuning techniques to improve models' capacities for adversarial robustness is an interesting one.
- The experimental validation is reasonably thorough (though there are issues in the setup).

**Weaknesses:**

This work currently has several weaknesses.

- There are several instances when the writing is imprecise. For instance
  - The claim made on lines 144-146 "Theoreticaly, Theorem 2.1 ... with Gaussian noised inputs" is a strong one, and requires either a citation or some kind of rigorous or empirical justification in the paper.
  - Lines 150-156 are also unclear. For instance, the authors should clarify "large-scale vision model ... acquire such potential ...". What does the phrase "acquire such potential" refer to?
  - lines 346-347: "this is mainly due ... especially for high resolution images" is another strong claim that either requires a citation or empirical justification
- It is unclear what the authors mean by "certified." Is it that the PEFT can be used to guarantee the conditions of Theorem 2.1? If so, can the authors provide a formal proof for this? In order to make a claim that a method is 'Certified', a rigorous guarantee must be provided.
- Equation (6) is written poorly and appears to be key to the entire experimental slate (and thus, this paper). Specifically:
  - The expression should be written in terms of indicator functions
  - The use of the '&' is unclear - do you mean the product?
  - The authors state that "certifiedCheck returns 1 if Theorem 2.1 is satisfied". However, the conditions in Theorem 2.1 are all probabilistic in nature. How are the probabilities computed? Specifically, how is $\mathrm{Pr}[F(x+\varepsilon) = c_A]$ computed?

**Questions:**

See 'Weaknesses' section for questions.

---

### Official Review · Reviewer_mJTE · 2024-11-04

**Soundness:** 3
**Presentation:** 2
**Contribution:** 3
**Rating:** 6
**Confidence:** 3

**Summary:**

In this paper, the authors propose a PEFT-inspired method for adapting pre-trained base models to produce robust classification performance under noise-augmented data. The authors propose a white-box method that augments a trainable layer to a given base model (of various architectures) and tunes its parameters with respect to a Cross-Entropy Loss under noise-augmented labeled data samples. They also provide a black-box approach that uses an AutoEncoder style Coordinator with a frozen encoder and a trainable decoder to denoise/augment the noisy data samples before feeding them to the base model. The black-box coordinator is trained using SPSA with Cross Entropy Loss under noise-augmented labeled data. Finally, the authors also propose a joint fine-tuning and robustifying training that retrains a large model to adapt it to a downstream task while also adapting it to the randomized smoothing noise injection.

The authors also provide extensive experiments for CIFAR-10 and Imagenet datasets, comparing the proposed methods to other SOTA denoised smoothing methods.

**Strengths:**

The proposed method combines the advantages of retraining and denoising-based randomized smoothing methods to provide a best-of-both-worlds solution. The black-box approach also makes PEFTSmooting easy to use with multiple different architectures without requiring further customization.

The experimental results of CIFAR10 beat the current SOTA methods by an impressively large margin.

The ablation studies, as well as the GradCAM experiments, are quite helpful for understanding the relative advantages of the proposed approach.

**Weaknesses:**

The presentation in the paper needs to improve. The paper assumes a lot of background on PEFT on the reader. Given that most of the results in the paper are based on LoRA, it might be useful to provide a detailed explanation of the setup and training of the LoRA model. Similarly, I would urge the authors to provide a more detailed overview of the four different PEFT approaches considered in the paper. Figure 1 does not adequately explain the PEFT setting.

The Imagenet results in the appendix are quite a lot worse compared to the Diffusion denoiser. While the authors suggest that this can be attributed mostly to the fact that the diffusion-based denoiser is more powerful for higher-resolution images, is there a smoother trade-off that can be established between the two models? A small discussion on this could be quite helpful.

**Questions:**

Please refer to the weaknesses section.

---

### Note · Authors · 2024-11-15

I have read and agree with the venue's withdrawal policy on behalf of myself and my co-authors.